# Mechanistic Insights into Palladium(II)-Catalyzed Carboxylation of Thiophene and Carbon Dioxide

Qingjun Zhang [1], Youguang Ma [1,2] and Aiwu Zeng [1,2,*]

1    State Key Laboratory of Chemical Engineering, School of Chemical Engineering and Technology, Tianjin University, Tianjin 300350, China; zqjlml2016@tju.edu.cn (Q.Z.); ygma@tju.edu.cn (Y.M.)

2    Chemical Engineering Research Center, Collaborative Innovative Center of Chemical Science and Engineering (Tianjin), Tianjin 300350, China

*    Correspondence: awzeng@tju.edu.cn; Tel.: +86-022-27404732; Fax: +86-022-27404496

**Abstract:** The mechanism in palladium-catalyzed carboxylation of thiophene and $CO_2$ is investigated using the density functional theory (DFT) calculations, including three consecutive steps of the formation of carbanion through breaking the C–H bond(s) via the palladium acetate, the elimination of acetic acid and the nucleophile attacking the weak electrophile $CO_2$ to form C–C bond. Results show that the C–C bond is formed through taking the three-membered cyclic conformation arrangement involving the interaction of the transition metal and the $CO_2$, and the $CO_2$ insertion step is the rate-determining step for this entire reaction process. Aiming to precisely disclose what factor determine the origin of the activation energy barrier in this carboxylation reaction, the distortion/interaction analysis is performed along with the entire reaction coordinate.

**Keywords:** transition metal; DFT; cyclic conformation; acyclic arrangement

## 1. Introduction

Direct carboxylation of aromatic compounds with $CO_2$ to produce the aromatic carboxylic acids is one of the most potent methods to realize the conversion of $CO_2$ into high-value chemicals [1–22]. This approach not only makes efficient use of arene and $CO_2$ resources but also overcomes the problems of multiple reaction steps, waste disposal and environmental pollution in the traditional synthesis pathways, while there are some challenges in realizing the C–H carboxylation under the moderate conditions due to the kinetically and thermodynamically stable $CO_2$ and less reactive aromatic compounds.

Among these aromatic substrates, achieving the direct carboxylation of thiophene is also an urgent problem since there is a large amount (12,500 t/y) of coking thiophene in the traditional refining process in China, while it has not been reasonably used. The vast majority of thiophene in the production process is simply removed or destroyed as the sulfur impurities through the pickling or hydrogenation strategies, resulting in the inefficient use of this resource and the significant environmental pollution. Additionally, thiophene is similar in nature to benzene and can be substituted for benzene in many applications. For example, thiophene-2-carboxylic acid and thiophene-2,5-dicarboxylic acid can be used to synthesize high-value polymers as a substitute for petrochemical derivatives benzoic acid and terephthalic acid. Taking the generation of 2-thiophenecarboxylic acid as an elucidating example, there are three mainly conventional synthesis paths, including the formylation [23], acetylation [24] and Grignard reagent [25] methods. These traditional synthesis pathways have some disadvantages, such as multiple reaction steps leading to the lower atomic efficiency, the application of toxic reagents and a large amount of waste discharges resulting in environmental pollution. As a result, a new feasible approach, direct carboxylation of thiophene with $CO_2$, should be necessarily designed from the dual perspectives of green synthesis and atomic efficiency, taking careful consideration of the

progress of the carbon fixation in recent years and the successful application of some aromatic arenes [5], such as toluene and phenol, carboxylation reactions.

The exploration of transition metal or transition-metal-complex-catalyzed carboxylation of aromatic compounds with $CO_2$ has received little attention [26–32]. Mizuno et al. [28] found that the direct carboxylation of pyridine derivatives (2-phenylpyridine) with $CO_2$ (1 atm) was achieved with the product yield of 73% under the joint action of transition metal rhodium complex and organoaluminium $AlMe_2(OMe)$ reagents. Lv et al. [29] investigated the corresponding mechanism of this Rh-catalyzed carboxylation process by taking the density functional theory (DFT) calculations. It mainly included the C–H bond oxidative addition, $CO_2$ insertion into Rh–C bond and metal transfer steps, wherein the $CO_2$ insertion step was the rate-controlling step in this reaction process. Meanwhile, the organoaluminium reagent acted as Lewis acid to facilitate the $CO_2$ insertion process. MSuga et al. [30] realized the direct carboxylation of benzene with $CO_2$ under transition metal rhodium complexes and organoaluminium reagents. Additionally, MSuga et al. [30] determined that the C–H bond cleavage is the rate-determining step for this carboxylation reaction via the intermolecular competitive kinetic isotope experiments. Lee et al. [31] reported that *tert*-butyl thiophene-2-carboxylate realized the direct carboxylation reaction with $CO_2$ under the joint action of silver salt, ligand and LiO*t*-Bu, with the product yield of 79%. Additionally, the $CO_2$ insertion step was the rate-determining step via the DFT calculations and kinetic isotope effect experiment. Fujiwara et al. [32] firstly realized the direct carboxylation of furan, thiophene, benzene and anisole aromatic compounds with $CO_2$ simply using the transition metal palladium (II) complex (palladium acetate) as a catalyst, as shown in Scheme 1, but its mechanism has not been investigated. As observed from the above analysis, the corresponding rate-determining step varies with the transition metal complexes in this carboxylation process. Nevertheless, there is something in common here, that is, the activation of C–H bond of aromatic compounds and the $CO_2$ inserting into the metal-nucleophile bond. Therefore, understanding the characteristics of the various steps in this reaction process is very necessary for the further development of this process. That is the purpose of this paper.

**Scheme 1.** Pd-catalyzed carboxylation of thiophene with $CO_2$.

Palladium-catalyzed carboxylation reaction can be anticipated to occur through the cleavage C–H bond of aromatic compound and $CO_2$ insertion into the Pd–C bond steps, as shown in Scheme 2 [32]. Nevertheless, there are still some crucial issues that have not been addressed in this proposed mechanism as follows: (a) What is the mode of proton abstraction? (b) What is the origin of the difference in energy barrier for the alternative C–H breaking modes? (c) Should the elimination of acetic acid be considered in the reaction mechanism? (d) What is the interaction mode between the transition metal and $CO_2$, and is there consistency? (e) Which step in this reaction process is the rate-determining step? (f) What are the variations in the interatomic interactions as the reaction proceeds? Therefore, the detailed mechanism of this palladium-catalyzed carboxylation reaction is studied via the DFT calculations, and the corresponding distortion/interaction–activation strain model analysis is employed to elucidate the factor controlling the origins of the height of the activation energy barrier for the different activation modes.

**Scheme 2.** Proposed reaction mechanism for the Pd(OAc)$_2$-promoted C–H carboxylation.

## 2. Results and Discussions

In this reaction model, the full molecular system is considered in the calculations without the truncations or any symmetry constraints. Palladium(II)-catalyzed carboxylation reaction of thiophene with $CO_2$ mainly consists of two consecutive steps, including that the formation of carbanion represented as the σ-palladium complex (metal-nucleophile bond) is the result of palladium(II) acetate induced deprotonation and the nucleophile attacks the weak electrophile $CO_2$ to form the C–C bond in the form of metal carboxylate. In the following two sections, these two steps are studied in detail.

### 2.1. C–H Activation Step

There are many modes for transition metal complexes to activate the C–H bond(s) of aromatic compounds owing to the influence of the valance state and the types of the metal, ligand structure(s) and auxiliary additives [33–35]. There have been two prevailing one-step C–H activation mechanisms, including the concerted metalation-deprotonation (CMD) and σ-metathesis. To determine the specific C–H bond activation mechanism in this reaction process, we have carried out DFT calculations on these two different C–H bond activation modes.

There are various coordination forms between the palladium Pd and the acetate anion ligand, and as it is well known, the lowest energy form of the transition metal complex, Pd being bound to the acetate through four oxygen atoms, is inactive. Therefore, this transition metal complex needs to be activated by dissociation of one or more coordination bonds in the complex to achieve the catalytic process. In this pre-activation step, a thermodynamically stable intermediate Reactant Complex_1 is formed with 0.66 kcal/mol endergonic when the palladium acetate combines with the thiophene, as shown by the blue curve in Figure 1. The metal palladium Pd binds to the ligand acetate through three oxygen atoms, and there is a weak interaction between the other dissociated oxygen atom and the proton to be broken on the thiophene ring; meanwhile, there is also a weak interaction between metal palladium and the carbon atom at the active site on the thiophene ring, which makes the proton on the thiophene ring easy to transfer to the acetate ligand. The correspondingly important geometry structure parameters are presented in Figure S1 of Supporting Information. The cleavage of the C–H bond is achieved by the transfer of proton to the acetate ligand through the highly ordered six-membered ring transition state TS_1, in which the cleavage of the C–H bond is carried out simultaneously with the formation of the Pd–C bond. In the structure TS_1, the C–H bond length in thiophene is elongated from 1.08 Å to 1.31 Å, and the distance between the metal palladium Pd and the carbanion is gradually reduced from 2.27 Å to 2.08 Å (shown in Figure S1). The activation free energy barrier of this deprotonation step is 10.59 kcal/mol when taking the free structure Reactant_1 of thiophene and palladium acetate as the benchmark. As the activated C–H bond distance gradually increases, the interactions between the proton acceptor and the proton donor gradually change from the weak interaction to the O–H chemical bond, and then the intermediate Int_1 of σ-Palladium complex containing acetic acid is formed. There is 2.16 kcal/mol endergonic for the reaction Reactant_1 → Int_1. By studying the structure of the intermediate Int_1, it is found that there has weak interaction between the hydrogen atom on the carboxyl group in the acetic acid structure and the thiophene ring, and the distance between the hydrogen atom and the carbon atom is 2.05 Å. Additionally, the metal palladium Pd is coordinated to the oxygen atom on the C=O in the acetic acid structure with a distance of 2.11 Å. Therefore, the removal of acetic acid requires 14.84 kcal/mol endergonic in the case of these two interactions. Furthermore, the necessity of the acetic acid removal in this reaction mechanism is explored in detail in the $CO_2$ insertion step.

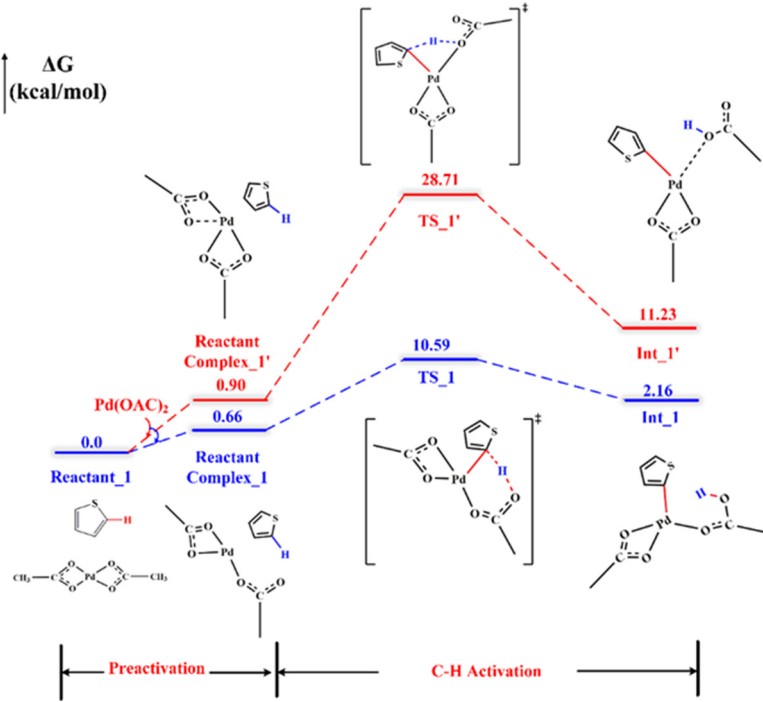

**Figure 1.** Reaction energy profiles of C–H bond cleavage step for CMD (blue curve) and σ-metathesis (red curve) modes.

The activation free energy barrier of the σ-metathesis deprotonation step is 28.71 kcal/mol when taking the free structure Reactant_1 of thiophene and palladium acetate as the benchmark, which is significantly higher than that of CMD mode (10.59 kcal/mol), where the corresponding origin for the difference in energy barrier under these two C–H bond breaking modes is shown in Section 2.4. As the activated C–H bond distance gradually increases, the intermediate Int_1′ of the σ-Palladium complex containing acetic acid is formed. There is 11.23 kcal/mol endergonic for the reaction Reactant_ 1→ Int_ 1′ , which is also greater than that of the CMD mechanism (2.16 kcal/mol). By studying the structure of the intermediate Int_1′, it is found that there is a weak interaction between the hydrogen atom on the carboxyl group in the acetic acid structure and the thiophene ring, and the distance between the hydrogen atom on the carboxyl group and the thiophene ring carbanion is 2.59 Å, which is larger than the corresponding interatomic distance (2.05 Å) in the CMD mechanism. In addition, the metal palladium Pd is coordinated to the oxygen atom on the OH in the acetic acid structure with a distance of 2.17 Å, which is also larger than the corresponding interatomic distance (2.11 Å) in the CMD mechanism. Thus, the interaction between the acetic acid structure and the complex is weaker in this mechanism in comparison with the CMD mode, so the energy required for the elimination of acetic acid in the σ-metathesis mode (5.77 kcal/mol) is less than that in the CMD mechanism (14.84 kcal/mol). Therefore, in this reaction process, the C–H activation takes the CMD mechanism.

### 2.2. $CO_2$ Insertion Step

The other key step for achieving this carboxylation reaction is the activation and polarization of the chemical inertness of $CO_2$ molecules. In this section, the influence of whether the acetic acid removal step is included in this palladium-catalyzed carboxylation of thiophene and $CO_2$ on the specific mechanism of subsequent nucleophile carbanion attacking weak electrophile $CO_2$ is discussed.

Figures 2 and S2 give the detailed mechanism of the nucleophile carbanion attacking the weak electrophile $CO_2$ and the correspondingly important geometric parameters in each complex structure in these two cases, respectively. The red curve in Figure 2 shows the detailed mechanism of carbanion attacking electrophile $CO_2$ when the acetic acid

removal step is not considered in the system. When the $CO_2$ molecule is close to the intermediate Int_1, the acetic acid structure is forced to twist due to the steric effect so that the intermediate can expose the carbanion center as much as possible. Note that the central atom of metal palladium Pd reaches coordination saturation with carboxylic acid anion, carbanion and acetic acid structure ligands so that there is no interaction between the metal palladium Pd and $CO_2$ molecule. The interaction of the nucleophile carbanion and the acetic acid structure with the $CO_2$ molecule leads to the formation of the stable intermediate Reactant Complex_2′ with 6.84 kcal/mol endergonic, and the detailed interatomic interaction isosurface maps are shown in Section 2.3. Under this interaction, the distance between the nucleophile carbanion and the $CO_2$ molecule is continuously shortened, forcing the bond angle of the $CO_2$ molecule to gradually become smaller and the molecule containing the Pd–C bond gradually moved away from its coplanar structure, resulting in the corresponding transition state structure TS_2′. The formed transition state structure TS_2′ mostly displays an acyclic conformation arrangement between the carbanion and the central carbon atom of $CO_2$; that is, there is no interaction between the metal palladium Pd and $CO_2$ molecule, but what really works is the carbanion. In the transition state TS_2′ structure, the distance between the central carbon atom of the $CO_2$ molecule and the carbanion of the thiophene ring is shortened from 3.23 Å to 1.82 Å, and the bond angle of the $CO_2$ molecule is bent from 178.30° to 140.78°. Additionally, the bond angle ∠C–Pd–C between the carbanion connected to the metal palladium Pd and the central carbon atom of $CO_2$ increases from 88.05° to 100.55°. The activation free energy barrier of this $CO_2$ insertion step is 43.09 kcal/mol when taking the free structure Reactant_1 of thiophene and palladium acetate as the benchmark, which is significantly higher than that of the deprotonation step. It also shows that it is very difficult to overcome such a high activation free energy barrier under the current reaction conditions. As the C–C distance between the carbanion and the central carbon atom of the $CO_2$ molecule continues to decrease, product complex Product Complex′ is formed. There is 38.04 kcal/mol endergonic for the conversion of $CO_2$ and intermediate Int_1 to the product complex Product Complex′. Additionally, the free energy barrier of the forward $CO_2$ insertion step is significantly larger than that of the dissociation process of the Product Complex′, which indicates that the product complex formed by this process is very thermodynamically and kinetically unstable. Therefore, it is necessary to consider the acetic acid elimination step in this reaction mechanism.

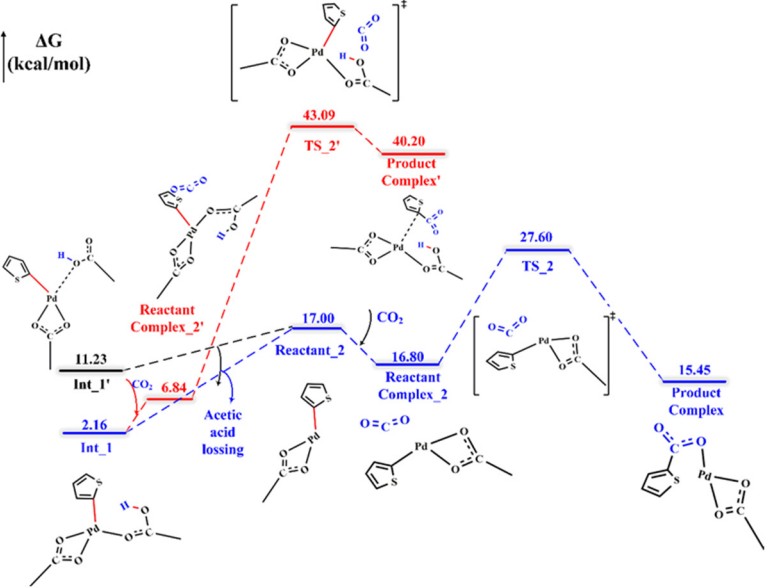

**Figure 2.** Reaction energy profiles of $CO_2$ insertion step: red curve for the case with the acetic acid; blue curve for the case with acetic acid removing.

Figure 2 (blue curve) gives the detailed mechanism of nucleophile carbanion attacking on electrophile $CO_2$ considering the acetic acid removal step in the system. The important geometry parameters of each complex structure are shown in Figure S2. The reactant intermediate Reactant_2 is obtained after eliminating acetic acid structure from the intermediate Int_1 (Int_1′) generated by the above proton abstraction step with 14.84 kcal/mol (5.77 kcal/mol) endergonic. The interaction of the carbanion and metal palladium with the $CO_2$ leads to the formation of the stable intermediate Reactant Complex_2 with 0.20 kcal/mol exoergic, and the detailed interatomic interaction isosurface maps are shown in Section 2.3. Under the joint action of metal palladium Pd, carbanion and $CO_2$ molecule, the distance between the nucleophile carbanion and $CO_2$ molecule is continuously shortened, forcing the bond angle of $CO_2$ molecule to gradually become smaller (from linear molecule to nonlinear molecule) and the molecule containing Pd–C bond gradually moved away from its coplanar structure, and the corresponding transition state structure TS_2 is obtained in this process. The formed TS_2 structure for C–C bond formation mostly displays the three-membered cyclic conformation arrangement, in which the oxygen and carbon atoms in the $CO_2$ molecule interact with the metal palladium and carbanion, respectively. In the TS_2 structure, the distance between the carbon atom of the $CO_2$ molecule and the carbanion of the thiophene ring is shortened from 3.10 Å to 2.01 Å, and the bond angle of the $CO_2$ molecule is bent from 177.08° to 145.39°. In addition, the bond angle ∠C–Pd–C of the carbanion connected to the metal palladium Pd and the central carbon atom of $CO_2$ is reduced from 71.73° to 48.55°, and the dihedral angle between the metal palladium Pd and the thiophene ring is changed from −178.73° to 156.11°. In other words, the metal palladium Pd is involved in both the activation of the $CO_2$ and the nucleophile, resulting in a strong interaction between the carbanion and the central carbon atom of $CO_2$. The activation free energy barrier of this $CO_2$ insertion step is 27.60 kcal/mol when taking the free structure Reactant_1 of substrate thiophene and palladium acetate as the benchmark, which is significantly smaller than that (43.09 kcal/mol) of the above process without considering the acetic acid removal step. As the C–C distance between the carbanion and the $CO_2$ molecule continues to decrease, a kinetically and thermodynamically stable product thiophene-2-carboxylate complex, Product Complex, is formed with 1.55 kcal/mol exergonic.

To summarize, it can be concluded that whether the reaction mechanism includes the acetic acid elimination step has a significant impact on the complex structure and interatomic interaction in the C–C bond formation process. First, there is a difference in the interaction mechanism between the transition metal M and $CO_2$ for these two cases. If considering the acetic acid removal step, the corresponding transition state structure mostly displays the three-membered cyclic conformation arrangement or inner-sphere complex, in which the oxygen and carbon atoms in the $CO_2$ molecule interact with the metal palladium and carbanion, respectively; otherwise, it is the acyclic conformation arrangement or outer-sphere complex, that is, there is no interaction between the transition metal M and the $CO_2$ molecule. Second, there are differences in activation free energy barriers for the $CO_2$ insertion step. The associated energy barrier of Pd–C bond insertion into $CO_2$ to form C–C bond is 43.09 kcal/mol if the acetic acid removal step is not taken into consideration, which is much higher than the energy barrier of 27.60 kcal/mol for another alternative. Third, there is a difference in the stability of the C–C bond formed by the $CO_2$ insertion step when considering the presence of acetic acid or not. Therefore, in this reaction process, the C–C bond is formed by taking the three-membered cyclic conformation arrangement involving the interaction of the transition metal palladium Pd and the $CO_2$ molecule. Additionally, the $CO_2$ insertion step is the rate-determining step for this entire reaction process.

### 2.3. $CO_2$ Interaction Modes

As observed from the above calculation results, the $CO_2$ insertion step is the rate-determining step in this palladium-catalyzed carboxylation reaction process. Furthermore, the interaction mechanism between the transition metal M and $CO_2$ molecules varies with whether the acetic acid elimination step is included in this reaction mechanism, that is,

three-membered cyclic conformation and acyclic arrangement. In order to determine the consistency of the interaction mechanism between transition metal M and $CO_2$ molecule during the carboxylation reaction, the effects of metal complexes generated by other transition metals, such as Cu(II), Ni(II), Rh(II) and Cu(I), and acetate ligand on the transition state structure and interatomic interaction formed by the nucleophile carbanion attacking the electrophile $CO_2$ molecule are also investigated.

Figure S6 shows the transition state structures of various metal complexes without the acetic acid removal step and their corresponding interatomic interaction isosurfaces. The key geometric parameters in the transition state structure, as shown in Table 1, and the IRI analysis results demonstrate that there is no interaction between transition metal M and $CO_2$ molecule in these four metal complexes, whereas $CO_2$ is activated through the joint interaction between the nucleophile carbanion and acetic acid in the system, which is consistent with the above findings in the investigation of metal palladium complex. Figure S7 shows the associated transition state structures of various metal complexes for the process involving the acetic acid removal step and their corresponding interatomic interaction isosurface. Table 1 and Figure S7 indicate that there is an obvious interaction between the transition metal M and the oxygen atom in the $CO_2$ molecule in these four metal complexes. There is also an electrostatic attraction between the transition metal M and the bent $CO_2$ molecule, demonstrating that the interaction of the transition metal M with the carbon and oxygen atoms in the $CO_2$ molecule activates the $CO_2$ molecule without coordination saturation or modest steric effect for the transition metal atom M, which is also consistent with the above findings in the investigation of metal palladium complex.

**Table 1.** The geometry parameters of each transition state structure for different transition metals.

| M | M-Complex | C–$O_A$ (Å) | C–$O_B$ (Å) | ∠O–C–O° | $C_{thio}$–$CO_2$ (Å) | M–$C_{thio}$ (Å) | M–$C_{CO2}$ (Å) | M–$O_{CO2}$ (Å) |
|---|---|---|---|---|---|---|---|---|
| Pd(II) | Complex with AA | 1.22 | 1.22 | 140.78 | 1.82 | 2.11 | 3.02 | 3.32 |
| | Complex without AA | 1.24 | 1.18 | 145.39 | 2.01 | 2.01 | 2.65 | 2.10 |
| Cu(II) | Complex with AA | 1.22 | 1.20 | 144.89 | 1.95 | 2.05 | 2.66 | 2.60 |
| | Complex without AA | 1.23 | 1.18 | 149.50 | 2.11 | 2.00 | 2.58 | 2.07 |
| Ni(II) | Complex with AA | 1.21 | 1.22 | 141.85 | 1.86 | 2.01 | 2.88 | 3.14 |
| | Complex without AA | 1.23 | 1.18 | 151.03 | 2.14 | 1.92 | 2.50 | 1.92 |
| Rh(II) | Complex with AA | 1.22 | 1.22 | 140.01 | 1.81 | 2.13 | 2.92 | 3.08 |
| | Complex without AA | 1.24 | 1.18 | 145.70 | 2.05 | 2.03 | 2.67 | 2.09 |
| Cu(I) | Complex with AA | 1.22 | 1.21 | 142.99 | 1.95 | 1.96 | 2.45 | 2.80 |
| | Complex without AA | 1.23 | 1.19 | 145.10 | 2.02 | 2.00 | 2.50 | 2.19 |

As a result, the transition state structure formed by the nucleophile carbanion attacking the $CO_2$ molecule is arranged in the form of a three-membered cyclic conformation or inner-sphere complex when the σ-metal complex formed in the deprotonation process does not reach coordination saturation or the steric hindrance effect around the central transition metal atom is small. The specific arrangement of the transition state structure is shown in Figure 3. The transition state structure formed in the $CO_2$ insertion step takes the form of an acyclic conformation or outer-sphere complex when the σ-metal complex formed in the deprotonation process reaches coordination saturation or the steric hindrance effect around the central transition metal atom is large. Figure 3 also depicts this unique $CO_2$ interaction mode.

## 2.4. Clarification of the Origin of Difference in Energy Barriers for Different Modes

Aiming to precisely disclose what factor caused the difference in activation energy barrier for these two proton abstraction pathways and two $CO_2$ interaction modes in this C–H carboxylation reaction proceed, the distortion/interaction-activation strain analysis model is employed [36–38]. The essence of this model is to decompose the relative energy $\Delta E(\zeta)$ of the potential energy surface into two contributions, including the activation strain energy $\Delta E_{strain}(\zeta)$, which is the energy required to twist the structures of the referenced two fragments, and the interaction energy $\Delta E_{int}(\zeta)$ between the two warped fragments along the reaction coordinate $\zeta$. Figure 4 gives the results of activation strain analyses for the

proton abstraction step of different C–H bond cleavage modes in the thiophene, along with the reaction coordinates. Depicted in Figure 4a illustrates that these two possibilities have the similar strain energy term along the reaction coordinate, while the higher activation energy barrier in the proton abstraction taking the σ-metathesis mechanism is stemmed from the least stabilizing interaction energy term between the substrate and the catalyst in the system when compared to the CMD mechanism, which will make the reaction more endothermic, which is consistent with the results of the above-mentioned analysis of the interaction within the complex structure using the IRI graphic method. Figure 4b shows the variation trend of the strain energy of each fragment in the system along the entire reaction process under the different C–H bond breaking modes. The deformation energy term for the substrate thiophene in the reaction process in the CMD or σ-metathesis mechanism is similar, but it is higher for palladium acetate, which is the largest contributor to the overall strain energy, as discovered. Therefore, the less activation energy barrier in CMD activation mode originates from the more stable interaction energy term in comparison with the σ-metathesis alternative, which leads to the transition state of the σ-metathesis pathway shifting to the later stage.

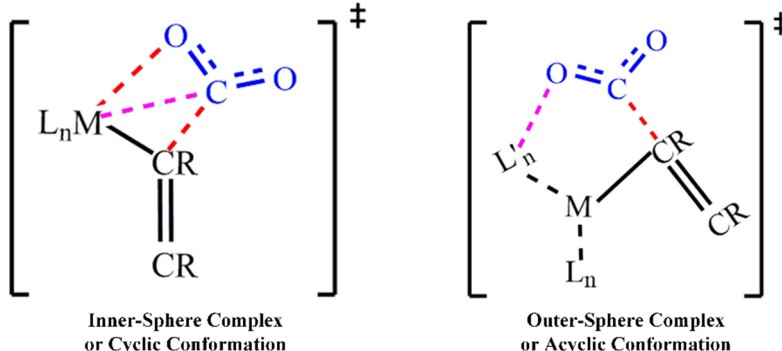

**Figure 3.** The possible interaction modes for the metal M and $CO_2$ in $CO_2$ insertion step.

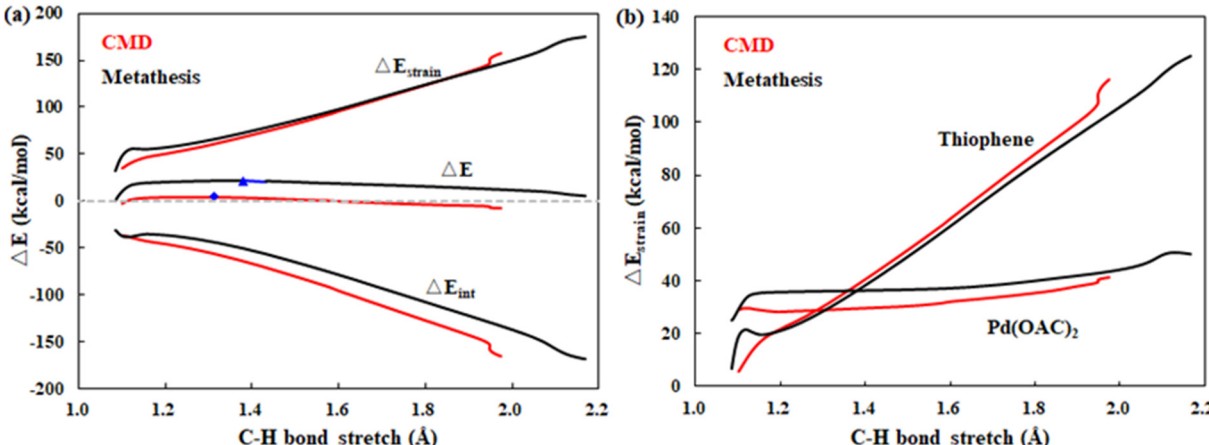

**Figure 4.** (**a**) Distortion/interaction model analyses for the alternative modes of deprotonation step along with the reaction coordinates; (**b**) the strain energy analyses for the deprotonation step along with the reaction coordinates for the thiophene and Pd(OAC)$_2$.

Furthermore, we also take the distortion/interaction-activation strain analysis model to analyze and determine the root for such a clear difference in the activation energy barrier in the system, allowing us to examine the influence of acetic acid more intuitively on the $CO_2$ insertion process. Figure 5 presents the corresponding analysis results for both cases. The difference in energy barrier is mostly due to the deformation energy term created by the structural distortion of metal palladium complex $\Delta E_{strain-Pd}$ (25.17 kcal/mol vs.

6.20 kcal/mol), although the deformation energy of $CO_2$ molecule is not much different (Figure 5). At the same time, there is a difference in the interaction between the twisted structures under the two pathways, with the inner-sphere complex pathway's interaction between $CO_2$ and the σ-palladium complex being stronger than the outer-sphere complex pathway's, which is consistent with the results of the above-mentioned analysis of the interaction within the complex structure using the IRI graphic method.

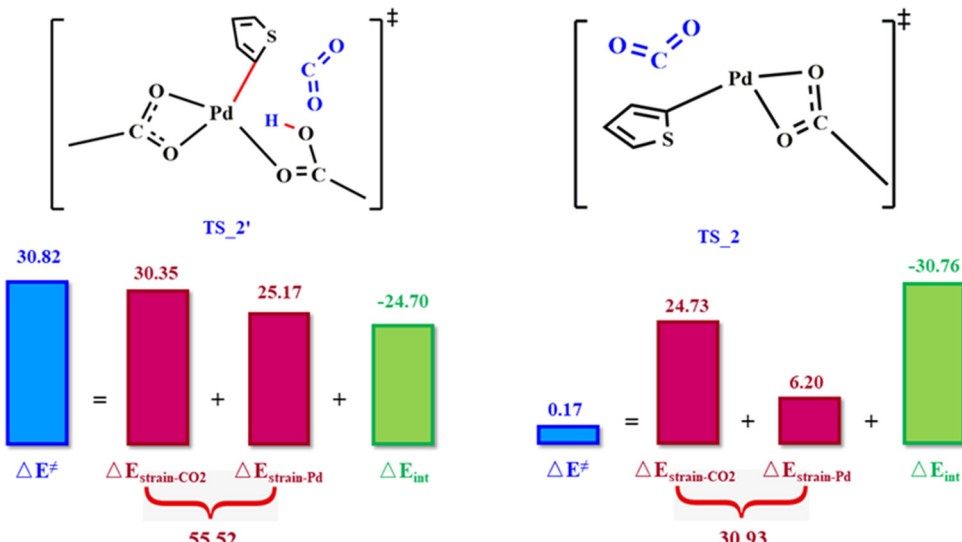

**Figure 5.** Distortion/interaction model analysis for the transition state structures TS_2 and TS_2′ of the $CO_2$ insertion step (the unit of the energy term is kcal/mol).

### 3. Computational Details

All the density functional theory (DFT) calculations were performed via the quantum chemistry package of Gaussian 16 (Rev.A03) suites [39]. Geometry optimizations for all referred structures of reactants or reactant complexes, intermediates and transition states, and products or product complexes were carried out at the B3LYP-D3(BJ) theory level with the mixed basis set including the Stuttgart effective core potential Lanl2dz [40] for all the transition metal atoms and 6–31G** for all other atoms in the implicit IEFPCM [41] solvent model. The combination of this theoretical method and the mixed basis set was named BS1. Frequency calculations were made to verify the nature of each structure, taking the same theory level and basis set as the geometry structure optimization. The characteristic of each transition state structure was evaluated by the intrinsic reaction coordinate (IRC) calculations, which can ensure that the transition state was connected to the correct reactants and intermediates. The electronic single-point energies were computed by the BS2 of the B2PLYP-D3(BJ) theory level with the def2-TZVP basis set for the structures obtained by the BS1. The corresponding solvation free energies were computed by the M05-2X theory level with the Stuttgart effective core potential SDD [42,43] for transition metal atom(s) and 6–31G* for all other atoms in the implicit SMD [44] solvent model. The combination of this theoretical method and the mixed basis set was named BS3. The final Gibbs free energies are computed as [45]

$$G = G_{BS1}^{corr} + E_{BS2} + E_{BS3}^{Solvent} - E_{BS3}^{Gas} + 1.89$$

### 4. Conclusions

In this paper, DFT calculations were performed to investigate the mechanism of the palladium acetate-catalyzed carboxylation of thiophene with $CO_2$. The reaction proceeded via the cleavage of the C–H bond, acetic acid removal and $CO_2$ insertion into the metal-nucleophile bond to give the aromatic thiophene carboxylate, wherein the $CO_2$ insertion step was the rate-determining step in the entire reaction process. The distortion/interaction-

activation strain analysis model revealed the origin of the difference in energy barrier for different interaction modes in the proton abstraction and $CO_2$ insertion steps, and results showed that the lower activation energy barrier in the CMD activation mode originated from the more stable interaction energy term in comparison with the σ-metathesis alternative, while the combined effect of strain energy and interaction energy terms determined the energy barrier in $CO_2$ insertion step. Note that whether the reaction mechanism included the acetic acid elimination step had a significant impact on the complex structure and interatomic interaction in the C–C bond formation process, especially for the $CO_2$ interaction modes, and their associated consistencies for $CO_2$ interaction modes were evaluated by the other transition metal complexes. Results showed that $CO_2$ was activated taking cyclic conformation under both the metal and nucleophile interactions when the σ-metal complex formed in the deprotonation process did not reach coordination saturation. Otherwise, it was taken by the acyclic arrangement. Additionally, for the purpose of deeply understanding the characteristics of this palladium-catalyzed C–H carboxylation reaction process, we visually analyzed the variations in the interatomic interactions as the reaction proceeded, taking the interaction region indicator analysis method, which simultaneously disclosed the changes in the noncovalent and covalent interactions along with the reaction coordinate.

**Supplementary Materials:** The following supporting information can be downloaded at: https://www.mdpi.com/article/10.3390/catal12060654/s1, Figure S1: Structures and important geometry parameters of complexes in C-H bond cleavage step for CMD and σ-metathesis modes.; Figure S2: Structures and important geometry parameters of complexes in $CO_2$ insertion step. Figure S3: The standard IRI color-bar and the chemical explanations of sign(λ2)ρ on IRI isosurface. Figure S4: The interaction analysis for each intermediate in C-H activation step for the CMD (a) and σ-metathesis (b) mechanisms. Figure S5: The interaction analysis for each intermediate in $CO_2$ insertion step. Figure S6: The TS geometry structure and the relevant interaction analyses including acetic acid for $CO_2$ insertion step: (a) $Cu(OAc)_2$; (b) $Cu(OAc)$; (c) $Ni(OAc)_2$; (d) $Rh(OAc)_2$. Figure S7: The TS geometry structure and the relevant interaction analyses excluding acetic acid for $CO_2$ insertion step: (a) $Cu(OAc)_2$; (b) $Cu(OAc)$; (c) $Ni(OAc)_2$; (d) $Rh(OAc)_2$. Figure S8: The interaction analysis for thiophene deprotonation step (CMD). Figure S9: The interaction analysis for the $CO_2$ insertion step [46,47].

**Author Contributions:** Data curation, Q.Z.; Formal analysis, Q.Z.; Investigation, Q.Z.; Project administration, A.Z.; Supervision, A.Z. and Y.M. All authors have read and agreed to the published version of the manuscript.

**Funding:** This research received no external funding.

**Data Availability Statement:** The data used to support the findings of this study are included within the article.

**Conflicts of Interest:** The authors declare no conflict of interest.

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
