# Peer review of "Mechanistic Insights into Palladium(II)-Catalyzed Carboxylation of Thiophene and Carbon Dioxide"

_catalysts, doi:10.3390/catal12060654_

Round 1

Reviewer 1 Report

The subject presented in this manuscript is currently important for the field of the catalytic processes.
The reaction mechanism is clearly presented, with scientifically arguments.

Author Response

Thanks for your comments and suggestions. We have made the revisions and corrections.

Reviewer 2 Report

In the case of tiophene, furan, cyclopentadiene and similar analogs of five-membered cyclic dienes, the mechasnism discussed can be competa with the DIels-Alder reaction involving one of C=O bond of carbon dioxide [Pure and Applied Chemistry, 93, 427 (2021)]. This should be mentioned in the introduction section.

"Concerted mechanisms" as well as "pericyclic rections" not exist [Eur. J. Org. Chem. 1107–1120 (2018)]. "Concerted" should be replaced to "one-step". 

The examples of the application of similar level of theory for resolving of similar problems should be included with respective references.

Figure 1,3:
Energetical profile should be include both dG and dH changes.

Probably, reactants complex is characterised by negative enthaply of the formation. Similar pre-reaction complexes were recently detected on the basis of DFT study regarding to different-type bimolecular processes. This should be mentioned  with respective references.

Author Response

Thanks for your comments and suggestions. We have made the corresponding revisions and corrections according to the Reviewer's comments. The relevant literature was cited in this manuscript. Besides,  for the reaction energy curves, the reviewer suggested that we add the enthalpy change of the reaction process. For this process, there is a limitation, that is, the parameters of the implicit solvent model are fitted against the experimental dissolution free energy under standard conditions, rather than the dissolution enthalpy, so the enthalpy change of this process can not be obtained. And the corresponding explanations are shown in the websites of Sobereva. http://sobereva.com/327.

Reviewer 3 Report

The article describes the study of the mechanisms of catalytic carboxylation of thiophene and carbon dioxide. Research is carried out using quantum-chemical calculations. In general, the work is written at a good level and well-constructed. The authors provide a good review of the literature regarding known carboxylation reactions using various catalysts. Thus, the relevance of the work is beyond doubt. Regarding the work itself, I would like to note that despite the interesting and important research topic, the article is very overloaded with various kinds of information and thus becomes difficult to read and perceive the information contained in it. In general, authors should review the content of the manuscript for the need for certain parts in its main text. Thus, some drawings containing information regarding geometric parameters can be transferred to the Electronic Accompanying Materials. Perhaps, at the discretion of the authors, some part of the textual material of the Results and Discussion part can also be moved there or simplified. Otherwise, the work turned out to be very interesting and made a deeply positive impression. The article should be published after minor corrections.

Author Response

Thanks for your comments and suggestions. We have made the corresponding revisions and corrections according to your comments. And the relevant descriptions and Figures are placed in the supporting materials. 

Round 2

Reviewer 2 Report

Authors considered all my remarks and improved the paper accordingly. So, i recommend the further evaluation of the manuscript.